# Efficient Multi-Threaded Data Starting Point Matching Method for Space Target Cataloging

**DOI:** 10.3390/s25082367

**Published:** 2025-04-08

**Authors:** Jiannan Sun, Zhe Kang, Zhenwei Li, Cunbo Fan

**Affiliations:** 1Changchun Observatory, National Astronomical Observatories, Chinese Academy of Sciences, Changchun 130117, China; sunjn@cho.ac.cn (J.S.); kangz@cho.ac.cn (Z.K.); lizw@cho.ac.cn (Z.L.); 2University of Chinese Academy of Sciences, Beijing 100049, China; 3Changchun Branch, Chinese Academy of Sciences, Changchun 130022, China

**Keywords:** optical data processing, orbit prediction, data matching, multi-threaded technology, space targets

## Abstract

Currently, multi-target survey telescope arrays play an important role in the build-up and maintenance of space object catalog databases, collecting massive observational data without attributing information. However, the matching process of massive observational data poses significant challenges to traditional prediction methods. To address the issues of low matching success rates and prolonged computation times in traditional methods, this paper proposes a multi-threaded data starting point matching method. First, orbital elements from the Space Surveillance and Tracking (SST) database are extracted for two days before and after the observation moment. A set of orbital elements closest to the observation epoch is filtered to form the primary candidate catalog containing the maximum number of objects. A matching error threshold is set. Second, multi-threaded traversal of the primary candidate catalog is performed to calculate observation residuals with the data starting point using an orbit prediction procedure. Orbital elements meeting the triple matching error threshold are selected to form the secondary candidate catalog, which is used in the entire data arc segment-matching calculation. Finally, the root mean square error (RMSE) of observation residuals for the entire data arc segment is computed point by point. The orbital elements satisfying the matching threshold are identified as matching results based on the principle of optimality. Experimental results demonstrate that with a matching error threshold of 1°, the proposed method achieves an average matching success rate of 97.62% for data arc segments with nearly 10,000 passes per day over 8 consecutive days. In the SST database containing an average of 25,720 targets, this method processes an average of 2164 data arc segments per minute, improving matching efficiency by 115 times compared to traditional prediction methods.

## 1. Introduction

The surge in space activities and the rising incidence of space collisions and disintegrations have resulted in a dramatic increase in the number of space targets. As of October 2024, the National Aeronautics and Space Administration (NASA) catalogs 28,658 space targets, marking an increase of approximately 1300 compared to the previous year [1]. The orbital parameters of these targets are in constant flux due to gravitational influences and various perturbative factors. Continuous observation is therefore required to accurately monitor their orbital characteristics and enhance space situational awareness. By obtaining precise positional information of space targets, effective maintenance and cataloging of these objects can be achieved. In recent years, sky surveys of space objects have emerged as a primary approach for cataloging [2,3,4]. In the realm of photoelectric detection, users can ascertain information such as the two-dimensional positional coordinates of space targets traversing the telescope’s field of view, along with their luminosity. In the field of radar detection, information on the azimuth, pitch angle, and slant range of space targets are obtained. However, when faced with multiple targets, these observational methods fall short in directly obtaining specific attribute information. To further utilize observational data, it is necessary to perform data matching and correlation operations on them.

The maintenance and establishment of the Space Surveillance and Tracking (SST) database hinge on effective matching and correlation of observations. In cataloging efforts, two primary methods of matching and correlation are employed: first, matching observational data with known entries in the catalog database; and second, correlating observational data across different datasets [5]. When a data center collects vast amounts of data from diverse observation platforms (both space-based and ground-based) and various types (optical and radar), the initial step typically involves matching the observational data with cataloged targets to update the orbital elements of catalog database through orbit determination methods. This step is crucial for maintaining the integrity of the catalog database. The subsequent correlation step involves linking unsuccessfully matched data to identify segments of data arcs belonging to the same target. This process aims to consolidate these segments for orbital improvement, ultimately determining whether a target is newly discovered based on comprehensive assessment and validation. However, the initial orbit determination used in this second correlation method can significantly affect the accuracy of orbit solutions, often resulting in errors on the order of tens to hundreds of kilometers [6,7,8,9]. Such inaccuracies can lead to erroneous associations of data arc segments, complicating the processing of uncorrelated target data.

Given the variability in the number of orbital elements cataloged daily and the inconsistent update frequencies of each target, the massive data matching process is susceptible to issues such as data leakage and misdetection. These challenges can adversely impact matching success rates and diminish overall efficiency, particularly due to the diverse matching methodologies employed. This paper primarily addresses the success rate and efficiency of matching observational data with the SST database. Wu [5] proposes a method that matches the time differences between observed data and known targets with the orbit surface difference sequence. This method achieves a correlation success rate up to 90%. However, some observation data still require initial orbit determination, and the parameters of two linear error sequences must be derived through robust estimation, complicating the computational process. Ding [10] and Lei [11] utilize observational data alongside two-line element (TLE) data for matching, attaining success rates of approximately 85% and 83%, respectively. Nevertheless, their research focused exclusively on low-orbit space objects and lacked comprehensive implementation strategies for catalog database extraction and target screening. Song [12] introduces an algorithm for associating targets in real-time geosynchronous orbit (GEO). This algorithm improves association rates in dense trajectory scenarios. However, it is limited by the types of observational data used. In addition, matching efficiency remains unaddressed in the above literature. Pastor [13] explores the generation of simulated radar data for known cataloged target tracks, correlating it with real measurement data, and concludes that this method offers a short computation time and effective correlation. Considering the nearly 30,000 known cataloged targets and the characteristics of extensive observational data, there is a pressing need to optimize the matching methods between observational data and known cataloged targets. This optimization is vital for rapidly enhancing space situational awareness capabilities and improving the cataloging and maintenance of space targets.

In response to these challenges, this paper proposes a multi-threaded data starting point matching method to enhance both the success and efficiency of matching massive observational data with orbital elements of the catalog database. The proper extraction of orbital elements of the catalog database is fundamental to achieving successful observational data matching and improving overall efficiency. On the one hand, multi-threading technology is used to quickly traverse and process the orbital elements of the catalog database. On the other hand, the data starting point matching method can constrain the number of targets entering the secondary candidate catalog by the observation residual at the data starting epoch. The root mean square error (RMSE) of the observation residuals is calculated point by point over the entire data arc. Ultimately, the matching result is based on optimality principles. This method demonstrates high matching success rates and efficiency. It can handle large volumes of optical and radar data, facilitating the matching association between observational data and known targets. This makes it suitable for managing and maintaining the SST database.

The structure of this paper is organized as follows: Section 2 outlines the principles of data matching and introduces the multi-threaded data starting point matching method, as well as the extraction principles of the orbital elements of the SST database; Section 3 presents the experimental results; Section 4 offers an insightful analysis of the findings and outlines potential directions for future development; and Section 5 contains the conclusion of the paper.

## 2. Materials and Methods

### 2.1. Data Matching Principles

The publicly available database of the North American Aerospace Defense Command (NORAD) serves as the orbital cataloging repository, with TLE (Two-Line Element) data released by NORAD continuously updated on a daily basis via the Space-Track website. The corresponding orbital calculation model, SGP4/SDP4 (Simplified General Perturbation Version 4/Simplified Deep-space Perturbation Version 4) [14], is employed to predict the orbital state, which represents in the True Equator and Mean Equinox of the date coordinate system (TEME), denoted as $r_{\mathrm{TEME}}$. For optical observations, the state vector derived from astronomical positioning data $\left(\alpha ,\delta \right)$ is referenced in the J2000 calendar year Mean Equator and Mean Equinox of date coordinate system. The state vector of the station position is defined in the Earth-Centered Earth-Fixed (ECEF) coordinate system, represented as $r_{b}$. When matching the observation data, it is essential to convert the TEME and ECEF coordinate systems to the J2000 coordinate system. The conversion relationship is expressed as follows [15,16]:(1)$$r_{J2000}={\left[M_{Z}\left(\Delta \psi \cos\varepsilon^{*}\right)\left(N\right)\left(A\right)\right]}^{T}r_{\mathrm{TEME}}$$(2)$$r_{J2000}={\left[\left(B_{2}\right)\left(B_{1}\right)\left(N\right)\left(A\right)\right]}^{T}r_{b}$$

Among them, $M_{Z}\left(\Delta \psi \cos\varepsilon^{*}\right)$ represents the rotation matrices by an angle $\Delta \psi \cos\varepsilon^{*}$ counterclockwise about the Z-axis, which Δψ indicates that the nutation in longitude, and $\varepsilon^{*}$ accounts for the true obliquity of the ecliptic. $B_{2}$ represents the polar motion correction matrix. $B_{1}$ is the earth rotation matrix. N is the nutation correction matrix. A is the precession correction matrix. M is the coordinate system rotation matrix. The superscript T indicates that the transpose of the transformation matrix. The parameters are defined as: α for right ascension, δ for declination.

Assuming that at the time of t, the position vector of a space target, as calculated by the orbit prediction model and transformed into the J2000 coordinate system, is denoted as $R_{t}\left(x_{t},y_{t},z_{t}\right)$. The position vector of the station in the same coordinate system is represented as $R_{st}\left(x_{st},y_{st},z_{st}\right)$. Then, the position vector of the space target in the J2000 coordinate system, centered at the station $R_{ct}\left(x_{ct},y_{ct},z_{ct}\right)$, is given by:(3)$$R_{ct}\left(x_{ct},y_{ct},z_{ct}\right)=R_{t}\left(x_{t},y_{t},z_{t}\right)-R_{st}\left(x_{st},y_{st},z_{st}\right)$$

The theoretical observation value of the space target at this moment is expressed as:(4)$$\left\{\begin{array}{l} \alpha_{ct}=\arctan\left(y_{ct}/x_{ct}\right)\\ \delta_{ct}=\arcsin\left(z_{ct}/\sqrt{x_{ct}^{2}+y_{ct}^{2}+z_{ct}^{2}}\right) \end{array}\right.$$

The observation residual for the observation $\left(\alpha_{ot},\delta_{ot}\right)$ is then calculated as:(5)$$\varepsilon =\sqrt{{\left[\left(\alpha_{ot}-\alpha_{ct}\right)\cdot \cos\delta_{ot}\right]}^{2}+{\left(\delta_{ot}-\delta_{ct}\right)}^{2}}$$

The traditional prediction matching method [5] operates as follows: It sets a matching error threshold and traverses the elements of the catalog database by the orbit prediction procedure. This generates a set of orbital elements visible during the observation period from the station. The target’s motion direction is derived from the unit vector of the observation value, which is utilized to filter the set of orbital elements participating in the observation arc segment matching based on the consistency of motion direction. Subsequently, the RMSE of the observation residuals for the data points within the observation arc is computed, and the matching result is determined according to the principle of optimality.

### 2.2. Multi-Threaded Data Starting Point Matching Methods

The primary factors influencing matching efficiency are the speed at which the orbital elements of the catalog database are traversed and the number of matching targets participating in the observation arc segment. To address these factors, this paper proposes a multi-threaded data starting point matching method. During data matching, each object in the primary candidate catalog needs to participate in data matching. Using multi-threaded parallel computing technology can improve traversal efficiency. More importantly, using the data starting point matching method, the vast majority of targets in the primary candidate catalog can be filtered out. This is because each target has only one calculation, after which it is determined whether it can enter the secondary candidate catalog. The relatively small number of targets in the secondary candidate catalog is the vital factor in improving the matching efficiency. The multi-threading technology and the data starting point matching method are complementary. Therefore, the multi-threaded data starting point matching method can boost the matching efficiency through two distinct approaches.

Multi-threading technology is a programming technique that enables programs to execute multiple threads simultaneously and is widely utilized in modern engineering applications [17]. Its core principles encompass the concepts of concurrency and parallelism. Concurrency refers to the interleaved execution of multiple threads within the same time frame, while parallelism involves the true simultaneous execution of multiple threads across multiple processors. Modern multi-core CPUs support parallel processing, significantly enhancing the efficiency of program execution. In this study, multi-threaded parallel programming [18] is implemented under Windows 7 using Visual Studio 2019 with C++. During the traversal of all orbital elements of the catalog database, multiple threads share the resources of the orbital elements, which can lead to data output contention and inconsistencies when matching the data starting point and performing point-by-point matching. To mitigate these issues, a synchronization mechanism (mutual exclusion locks) is employed to manage data storage and ensure thread safety.

Assuming that the number of space targets over the station is uniformly distributed across the base surface of the cone $S_{1}$ at a zenith distance of $\theta_{1}$ (see Figure 1). The unit vector of the observation data of the space targets at a given moment is denoted as $\hat{L}={\left[\cos\delta \cos\alpha ,\cos\delta \sin\alpha ,\sin\delta \right]}^{T}$. The base surface $S_{2}$ represents the number of matching targets within the observation arc, while $\theta_{2}$ is the angle between the cone’s axis and the unit vector of the starting point of the observation arc when the space target intersects the base surface. The angle between the unit vector of the starting point of the observation arc and the direction vector p of the space target is used to determine whether the space target enters the base surface, denoted as $\theta =\arccos\left(\hat{L}\cdot p/\left|\hat{L}\right|\left|p\right|\right)\cdot \left(180/\pi \right)\leq \theta_{2}$. After performing a simple calculation, we can derive the relationship between these variables, presented as $S_{1}/S_{2}=\tan^{2}\left(\theta_{1}\right)/\tan^{2}\left(\theta_{2}\right)$.

Due to $S_{2}=\pi h^{2}\tan^{2}\left(\theta_{2}\right)$, its size directly impacts the number of participating space targets matched in the observation arc segment, which is proportional to the square of h and $\tan\left(\theta_{2}\right)$ to ensure applicability to the matching processing of target observation data with different orbital altitudes without the need to qualify h. This paper employs a triple matching error threshold to control $\theta_{2}$. If the number of compute system threads is $N_{p}$, the calculation efficiency in the multi-threaded data starting point matching method is proportional to both $N_{p}$ and $\tan^{2}\left(\theta_{1}\right)/\tan^{2}\left(\theta_{2}\right)$. In the traditional prediction matching method, $\theta_{1}$ is set to 75°, while in the data starting point matching method, $\theta_{2}$ is reduced to 3°. By substituting the relevant parameters into the above relationship, theoretical calculation demonstrates that the proposed method achieves an efficiency improvement of several orders of magnitude compared to traditional approaches.

In terms of processing efficiency, the proposed method enhances upon traditional approaches through two key innovations: First, it employs parallelized multi-threading for catalog database traversal, improving traversal efficiency. Second, it eliminates the requirement for predicting visible objects or verifying motion direction consistency. Instead, it directly utilizes the tracking data at the data starting point epoch to compare against theoretical value derived from orbital propagation. This strategy restricts the number of objects participating in data arc segment calculations via a threefold matching error threshold. Thereby, matching efficiency has been greatly improved.

### 2.3. Extraction of Orbital Elements for Catalog Database

In the process of matching observation data with the NORAD catalog database, the proper extraction of TLE data is crucial, serving as the foundation for enhancing both the matching success rate and efficiency. The extraction of TLE data from the NORAD catalog involves three critical considerations: The first is the number of days covered by the TLE data. The longer the time span, the more objects the NORAD catalog contains. However, this will consume substantial reading time and affect matching efficiency. A balance should be struck between the time span and reading time. The second is the elimination of abnormal data. Incorrect orbital elements also affect matching efficiency. The third is the selection of multiple sets of orbital elements for the same object. The set of orbital elements closest to the epoch of observation should be selected to reduce orbit propagation error.

The published TLE data often contain anomalies or outlier values that must be eliminated prior to matching. The criteria for identifying outlier values [19] are based on the following: the product of the changes in the semi-major axis of the neighboring data points is negative, and the sum of these changes is small.

As an example, we utilize the TLE data published on the Space-Track website from 1–30 September 2024 [20]. Figure 2 illustrates the relationship between the number of targets included in the TLE data across different time spans, ranging from a single day to six days. The TLE data selection rule involves extracting data for half of the consecutive days both before and after the reference time, while counting the number of unique targets that do not appear repeatedly.

For the TLE data extracted over spans from one to six days, the average total number of targets over a period of 30 consecutive days is calculated. The growth rates of the number of targets are found to be as follows: 6.90%, 2.46%, 1.33%, 0.82%, and 0.59%.

While extracting TLE data over multiple days will undoubtedly increase the total number of included targets, it also leads to excessive computer reading time, thereby reducing matching efficiency. Moreover, once the matching success rate reaches a certain threshold, further increases in the total number of targets will not yield proportional benefits to the success rate. For targets with multiple sets of TLE data, failing to select the data set closest to the observation moment can adversely affect the matching success rate due to the propagation of orbit prediction errors.

Based on the aforementioned principles, Figure 3 presents the flowchart of the data starting point matching method utilizing multi-threading techniques.

## 3. Experimental Results

To validate the matching success rate and computational efficiency of the proposed TLE selection criteria and multi-threaded data starting point matching method, experiments were conducted using observation datasets from the Jilin Astronomical Observatory. The telescope parameters and specifications are detailed in Table 1. The computer hardware configuration includes an i7-4790 CPU running at 3.6 GHz with 4 cores and 8 threads, along with 24.0 GB of RAM.

Observations are conducted from 21–28 September 2024. A cooperative target captured during this period is selected to evaluate the measurement accuracy of the telescope using its consolidated prediction format (CPF) ephemeris, which serves to verify the stability of the telescope’s operational conditions, as shown in Table 2.

During the observation period, the measurement accuracy of the telescope is assessed for the Jason-3 satellite, which has an orbital altitude of 1336 km. According to Table 2, the RMSE in the direction of right ascension is 2.88″, the RMSE in the direction of declination is 2.84″, and the total RMSE is 4.03″. The results indicate that the telescope’s measurement accuracy is indeed stable. This level of accuracy is essential for conducting precise astronomical observations and demonstrates the telescope’s effective performance during the measurement period. Such stability in measurement accuracy is crucial for ensuring successful matching with TLE data and enhances the overall quality of observational data.

Matching experiments are conducted on the TLE data spanning from a single day to six days based on the above observations, respectively, utilizing a multi-threaded data starting point approach with the matching threshold set to 1°. By analyzing the average matching success rate and efficiency of the observed data over an 8-day period, as shown in Figure 4, we establish a clear relationship between the number of days spanned by the TLE data and the effectiveness of the matching process. The calculations reveal that adjusting the time span of the TLE data directly influenced both the matching success rate and the processing time.

Figure 4a shows that as the number of days of TLE data span increases, the average matching time consumed and the matching success rate for processing each 10,000 passes of data increase. Figure 4b shows the variability of the matching time consumed and the matching success rate. The variability of matching time reaches its minimum when the catalog database has 4 days of TLE data. At this time, the variability of the matching success rate is close to zero.

The TLE database with a 4-day span is selected for matching. The total number of targets in the catalog database per day is presented in Table 3. The 8-day observation data are processed using the multi-threaded data starting point method. The matching error threshold is set to 1°. The parameter $\theta_{2}$ controlling the number of matched targets in the participating data arcs is set to 1°, 3°, 5°, and 10°, respectively. The average matching success rate over the 8 days is plotted against the average matching time consumed per 10,000 passes of data processing, as shown in Figure 5. Additionally, the matching error threshold is set to 60″, 360″, 1800″, and 3600″, with the parameter $\theta_{2}$ set to 3°. The average matching success rate over the 8 days is again plotted against the average matching time consumed per 10,000 passes of data processing, as illustrated in Figure 6.

From Figure 5, it is observed that when the matching error threshold is set to 1°, increasing the value of the parameter $\theta_{2}$ leads to an increase in both the matching time consumed and the matching success rate for every 10,000 passes of data processed. Specifically, when the parameter $\theta_{2}$ changes from 1° to 3°, the matching time increases by only 0.1 s, while the matching success rate improves by 1%. However, as the parameter $\theta_{2}$ continues to increase, the matching time increases more significantly, while the improvement in the matching success rate diminishes.

As can be seen in Figure 6, when the parameter $\theta_{2}$ is set to 3°, the average matching time consumed for processing each 10,000 passes of data increases only slightly as the matching error threshold rises. However, the matching success rate increases substantially from 40.56% to 97.62%, indicating a significant improvement. This suggests that the value of the matching error threshold is a critical factor influencing the matching success rate.

In summary, extracting TLE data spanning 4 days for observations with a matching error threshold set to 1° results in a better balance between matching efficiency and success rate when the parameter $\theta_{2}$ is set to 3°.

To evaluate the computational efficiency of the proposed multi-threaded data starting point matching method, we compare it against the traditional prediction matching method, multi-threaded matching method, and data starting point matching method. With the preceding selected parameters, the 8-day observational dataset was processed using the four methods described above. The matching time consumed is summarized in Table 4.

Analysis of the 8-day observational dataset demonstrates significant computational efficiency disparities among the four matching methods. For a catalog database containing an average of 25,720 space targets per day, the traditional method requires 31,885.76 s (8.86 h) to process 10,000 observation passes. The multi-threaded approach reduces this to 14,788.87 s (4.11 h), achieving a 2.15 × speedup. Notably, the data starting point matching method yields a dramatic improvement, processing the same dataset in 845.91 s (14.1 min)—a 37.7 × efficiency gain over the traditional prediction method. The integration of multi-threading and data starting point matching method further optimizes computational efficiency, yielding a processing time of 277.19 s (4.62 min). This represents a 115.0 × speedup relative to the traditional prediction method, signifying a two-order-of-magnitude enhancement in computational efficiency. Such improvements not only expedite the matching pipeline but also enable real-time analysis and decision-making in observational astronomy. Thereby, the proposed multi-threaded data starting point matching method demonstrates high applicability for large-scale observation data processing.

For TLE data spanning 4 days, various extraction methods are evaluated to confirm the high matching success rate associated with selecting the set of elements closest to the observation moment as the primary matching catalog database. The methodology involves extracting the orbital elements at the first appearance of the target, as well as the orbital elements of the nearest release ephemeris relative to the observation moment. The results of this validation process are illustrated in Figure 7, which compares the data matching success rate and time consumed by different extraction methods over the 8-day observation period. The average matching success rate achieved with the orbital elements’ first appearance extraction method is 77.78%, accompanied by an average processing time of 287.32 s. In contrast, the extraction method proposed in this study demonstrates a significantly higher average matching success rate of 97.62%, while maintaining a slightly lower average time consumption of 277.19 s. This indicates that the proposed method not only enhances the matching success rate but also improves processing efficiency, thereby providing a more effective approach for TLE data extraction and matching in observational astronomy.

## 4. Discussion

The proposed multi-threaded data starting point matching method demonstrates significant advancements in both efficiency and matching success rate for observation data processing. Compared to traditional prediction methods, which rely on the consistency of movement direction and broad error thresholds, our approach achieves a 115-fold efficiency improvement while maintaining a 97.62% matching success rate under a 1° matching error threshold. This significant improvement in performance can be attributed to two key innovations: (1) By selecting TLE data that is closest in time to the observation data epoch, the impact of orbit propagation errors is minimized, which enhances the success rate of matching. (2) The use of multi-threaded parallel computing technology increases the efficiency of catalog traversal. It is noteworthy that the data starting point matching method plays a crucial role. Leveraging the observation residual of the data starting point can effectively limit the number of targets participating in the full-arc calculation. Thereby, the matching efficiency is improved.

As shown in Table 5, our method outperforms existing approaches in both efficiency and applicability. For instance, Wu’s method [5] has a matching success rate of up to 90%, but it requires computationally intensive robust estimation and is complicated to operate. Ding’s method [10] reports an 85% matching success rate for low-orbit targets, but omits the high-orbit targets validation. Lei’s method [11] reaches a matching success rate of 83.64%. In the above literature, there is no mention of how to select the TLE data for the same target. This is crucial for influencing the matching success rate.

The results suggest that a 4-day span of TLE data provides the best compromise, where the balance between matching efficiency and success rate is achieved. By adjusting the parameter $\theta_{2}$, the number of targets involved in the full arc calculation is significantly reduced. The smaller the value of the parameter $\theta_{2}$, the less time the matching process requires. However, this results in a lower matching success rate (Figure 5). Similarly, a greater matching threshold leads to a higher matching success rate. However, once it reaches a certain value, further increases in the parameter will not result in additional increases in the matching success rate (Figure 6). Therefore, we recommend setting the matching error threshold to 1° and the parameter $\theta_{2}$ to 3° to achieve optimal matching results.

In this paper, we examine the issues of matching success and matching efficiency. Obviously, a smaller matching error threshold guarantees correct object matching but reduces the overall matching success rate and affects data utilization. Conversely, a larger matching error threshold, according to the principle of optimality, encompasses the correct results obtained under smaller thresholds. However, results derived from larger thresholds may be susceptible to fuzzy matching effects. Therefore, further research will also concentrate on fuzzy matching to enhance the correct matching rate.

Looking ahead, the exponential growth of space targets necessitates advanced data processing solutions. The multi-threaded data starting point matching method proposed in this study demonstrates strong potential as a pivotal tool for large-scale astronomical data processing. Given its computational efficiency, this method could be implemented as a dynamic-link library (DLL) and integrated into the astronomical positioning processing of the array telescope. Such integration would enable automated embedding of target attribute information (e.g., orbital parameters, identification codes) into observational data streams, significantly streamlining catalog management and cross-mission target recognition. Future research will focus on optimizing the algorithm’s real-time performance and enhancing its adaptability to various data processing scenarios, ensuring it remains effective in an evolving landscape of astronomical data.

## 5. Conclusions

This paper presents a multi-threaded data starting point matching method, showcasing its advantages in large-scale data processing. The experimental results affirm the method’s effectiveness, particularly in achieving a balance between matching success rate and efficiency.

For optimal performance, it is recommended to select TLE data spanning 4 days as the catalog database for matching. The matching error threshold is set to 1°, and the parameter controlling the number of participating arc segment matching targets is set to 3°. In terms of matching success rate, we select a set of orbital elements that are closest to the observational data epoch. This approach can effectively mitigate the influence of orbital propagation errors, thereby improving the matching success rate. Regarding matching efficiency, we adopt the multi-threaded technique and the data starting point matching method. In the latter method, it is determined by the observation residual at the data starting point whether a target will participate in full-arc segment matching. This method notably reduces the number of entries that need to be processed in the secondary candidate catalog. This strategic approach leads to a remarkable improvement in matching efficiency, achieving a performance that is two orders of magnitude greater than that of traditional prediction matching methods.

## Figures and Tables

**Figure 1 sensors-25-02367-f001:**
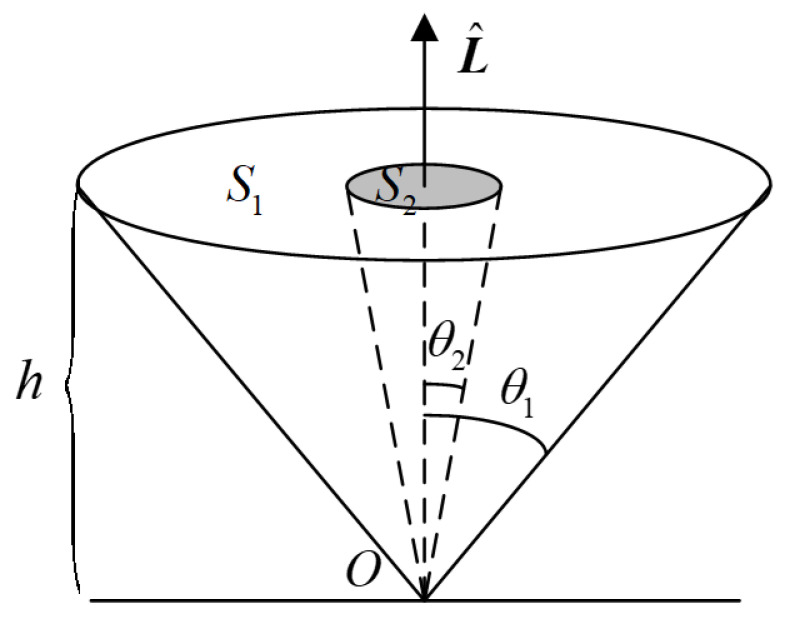
Schematic of the number of matching targets in the traditional prediction method and the multi-threaded data starting point method.

**Figure 2 sensors-25-02367-f002:**
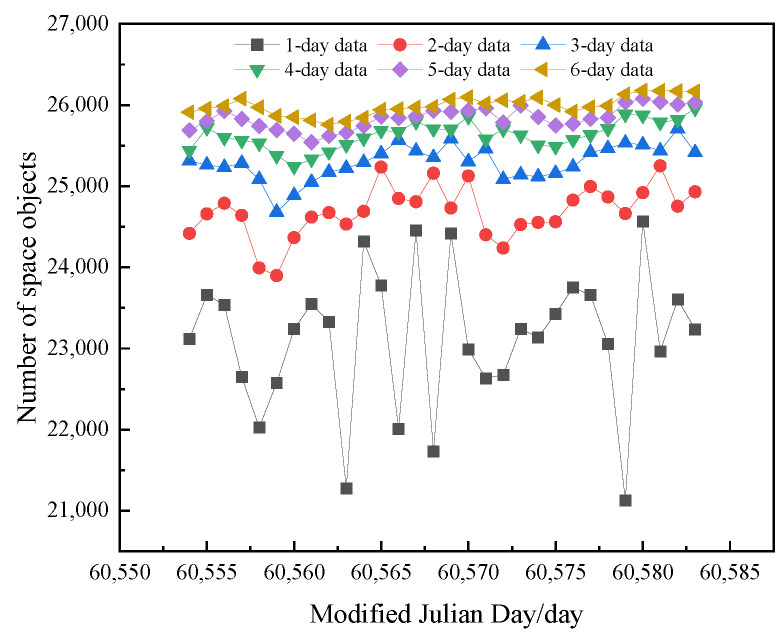
Number of targets included in TLE data for varying spans of days.

**Figure 3 sensors-25-02367-f003:**
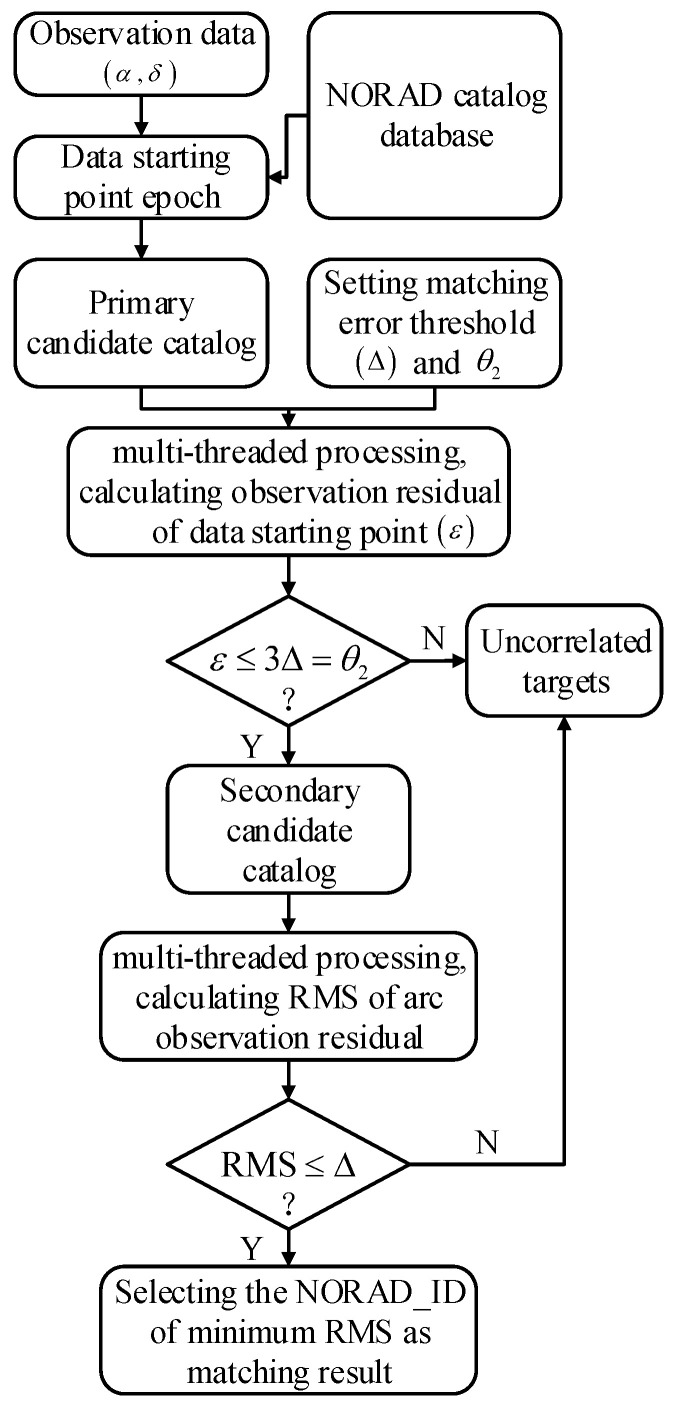
Flowchart of data starting point matching method with multi-threading techniques.

**Figure 4 sensors-25-02367-f004:**
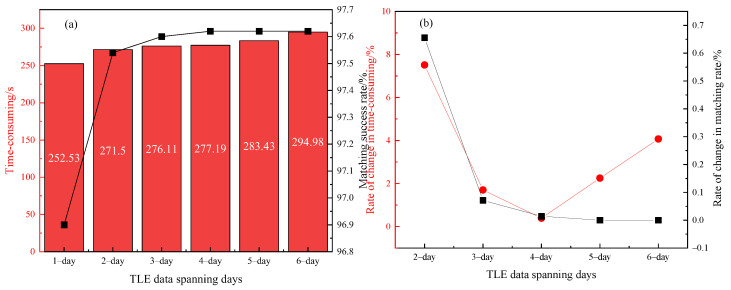
(**a**) Time consumed and matching success rate for different TLE data spanning days; (**b**) Variation rate of time consumed and matching success rate in (**a**).

**Figure 5 sensors-25-02367-f005:**
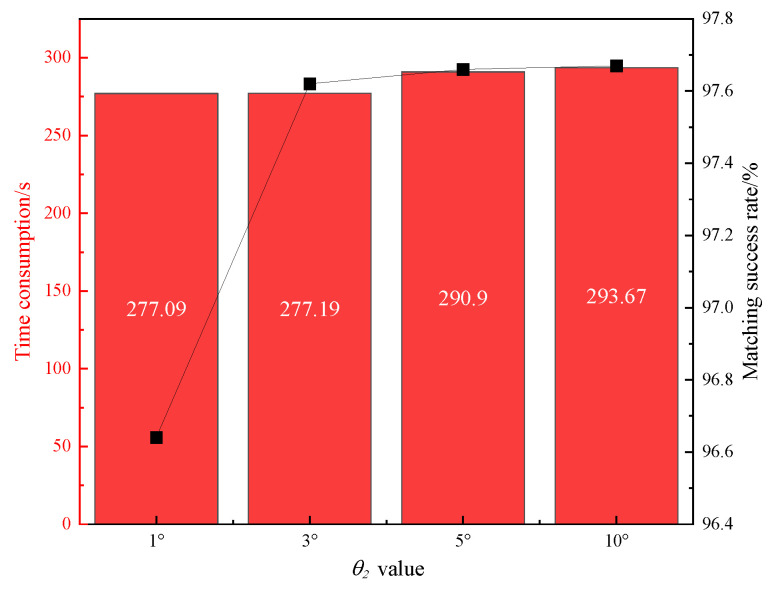
Time consumed and matching success rate for different $\theta_{2}$.

**Figure 6 sensors-25-02367-f006:**
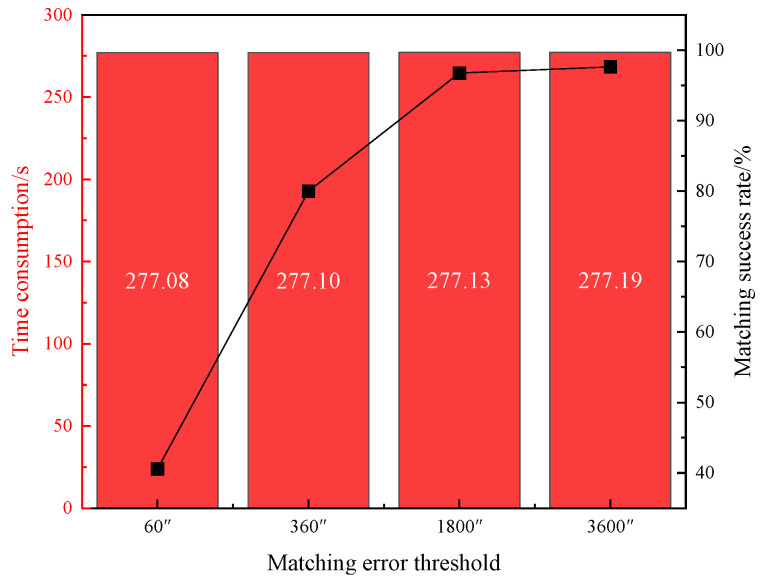
Time consumed and matching success rate for different matching error threshold.

**Figure 7 sensors-25-02367-f007:**
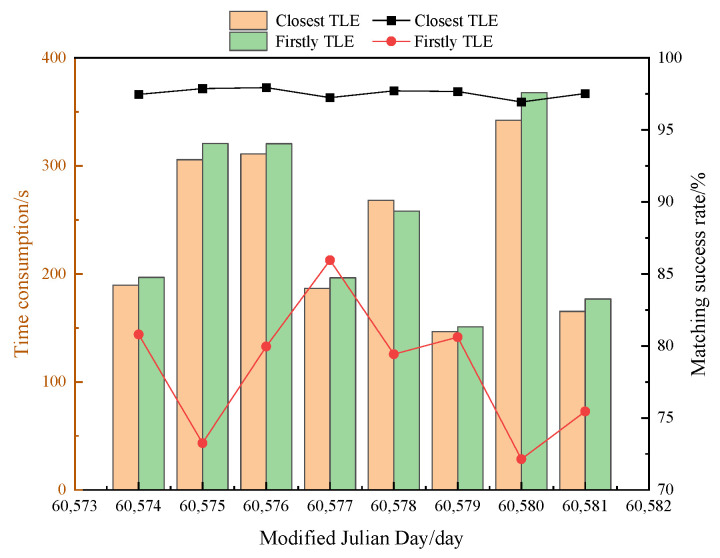
Time consumed and matching success rate for different TLE data extraction methods.

**Table 1 sensors-25-02367-t001:** Technical parameters of the telescope.

Parameters	Value
Monitoring region/deg^2^	1590
Resolution of CCD	3056 × 3056
Pixel size/μm	12
Observation accuracy/(″)	Piror to 9
Detection ability/mag	10.5

**Table 2 sensors-25-02367-t002:** Measurement accuracy assessment of Jason-3 satellite.

NORAD_ID	Observation Date	Arc Length/s	RMS_Right Ascension/(″)	RMS_ Declination/(″)	RMS/(″)
41240	2024-09-21 18:48:44.96	97.74	2.59	3.01	3.97
41240	2024-09-22 17:14:54.97	135.54	2.52	1.99	3.21
41240	2024-09-23 17:37:42.25	128.31	3.18	2.58	4.10
41240	2024-09-24 18:00:50.89	92.17	2.47	3.56	4.33
41240	2024-09-25 16:26:59.23	121.08	2.73	2.39	3.63
41240	2024-09-27 17:12:17.93	126.51	3.52	2.97	4.50
41240	2024-09-28 15:40:06.97	41.57	2.97	3.12	4.31

**Table 3 sensors-25-02367-t003:** Number of space targets included in TLE data spanning 4 days.

Date	21 September	22 September	23 September	24 September	25 September	26 September	27 September	28 September
Numbers	25,488	25,572	25,634	25,709	25,885	25,873	25,786	25,817

**Table 4 sensors-25-02367-t004:** Comparison of time consumption of four matching methods.

Date	Number of Observation Data/Pass	Time Consumption/Second			
		Traditional Prediction Method	Multi-Thread Method	Data Starting Point Method	Multi-Threaded Data Starting Point Method
21 September 2024	7227	23,054.69	10,694.82	608.90	193.27
22 September 2024	11,735	36,997.58	17,255.81	982.39	320.65
23 September 2024	11,788	37,869.85	17,450.91	994.00	326.78
24 September 2024	6993	22,410.69	10,355.91	596.10	198.01
25 September 2024	9362	29,662.66	13,860.05	797.05	260.06
26 September 2024	5488	17,640.63	8127.84	467.53	157.35
27 September 2024	13,023	41,896.62	19,281.57	1106.59	359.78
28 September 2024	6147	19,289.05	9102.47	517.96	173.33

**Table 5 sensors-25-02367-t005:** Performance comparison with existing methods.

Method	Matching Success Rate (%)	Efficiency (Second/10k Arcs)
Wu (2011) [5]	90	31,885.76
Ding (2019) [10]	85	Not mentioned
Lei (2019) [11]	83.64	Not mentioned
Proposed method	97.62	277.19

## Data Availability

The data presented in this study are available on request from the corresponding author. The data are not publicly available due to privacy restrictions.

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
