# Peer review of "Efficient Multi-Threaded Data Starting Point Matching Method for Space Target Cataloging"

_sensors, 2025, doi:10.3390/s25082367_

Round 1
Reviewer 1 Report
Comments and Suggestions for Authors
General Comments:
The manuscript addresses the important problem of space object observation-to-object matching. The authors propose a multi-threading technique based on the starting point of data, achieving high success rate and efficiency in matching massive observational arcs with current space object catalog. The proposed method appears robust and innovative, with sufficient technical details and systematic presentation. The whole paper is well-organized and the ideas are clearly articulated. In my opinion, the manuscript fits well with the scope of Sensors.
However, I have several concerns that need to be addressed:
- In the Introduction, Page 3, line 92, spatial situational awareness should be revised to space situational awareness.
- The primary cataloging library and secondary cataloging library should be clearly defined, and their role in the matching process should be emphasized.
- Clarification of Techniques in Section 2: In this section, a multi-threaded data starting pointing matching method is proposed. I am confused by this expression, as the multi-threading technology is a programming technique, while data starting pointing matching is a data processing technique. The relationship between these two techniques in the proposed method should be clearly explained? More importantly, it is essential to clarify which technique dominates the matching of massive arcs in the proposed method.
- In Section 2.2, the development of the proposed method is too brief. The key contributions of the method, such as those illustrated in Figure 1 and related mathematical expressions, are not clearly articulated. More detailed explanations are needed to clarify the proposed method and highlight its innovative aspects.
- In Section 2.3, the extraction of TLE data from the NORAD catalog is described based on a self-defined principle, which appears overly simplistic and vague. More details about the extraction process should be provided, including a clearly defined extraction principle, to ensure the reliability and reproducibility of the method.
- The orbital altitude of Jason-3 remains constant at 1336 km. This information can be included in the context or the table title rather than within the table itself. Additionally, in the title of Table 2, "Jason3" should be corrected to "Jason-3."
- In the figures, the term "matching rate" should be revised to "matching success rate" for clarity and consistency.
- The matching error threshold is a critical factor affecting the matching success rate. According to the matching rule, the TLE prediction error is included in the theoretical value. Since TLE prediction accuracy varies for space objects at different orbital altitudes, a single matching error threshold may not be sufficient or rigorous. The authors should provide a detailed explanation of how the matching error threshold is set and justified.
- The experimental results demonstrate excellent matching efficiency and a high matching success rate with a large amount of tracking data, compared to traditional matching methods. However, it is unclear whether this improvement primarily results from the use of the multi-threading technique or the data control in the secondary-level matching catalog. The authors should clarify the contributions of each aspect to the overall performance improvement.
The English could be improved to more clearly express the research.
Reviewer 2 Report
Comments and Suggestions for Authors
The paper presents a matching method that utilizes a multi-threading technique based on the starting point of data, showcasing its advantages in large-scale data processing. The experimental results affirm the method's effectiveness, particularly in achieving a balance between matching success rate and efficiency. This approach demonstrates significant applicability in space debris surveillance, cataloging, and related operations.
P4, Line 151-152: Why was spherical angular distance not used when evaluating the residuals of observational data? Equation (6) appears to be a small-angle approximation of spherical angular distance. Can its accuracy meet the requirements of all observational scenarios?
P3, section 2.1: In this section, you mentioned radar data, but the paper does not describe any radar observation experiments. In practice, radar observations can provide range information. Since your method does not utilize range information, does it perform as effectively in radar data association as it does with optical data? I recommend avoiding any mention of radar-related content.
Reviewer 3 Report
Comments and Suggestions for Authors
The main premises of the manuscript are not correct. For instance, the statement that the build-up and maintenance of space objects catalogues does primarily rely on survey telescope array is not correct. It depends on the particular sensor network and catalogue, but tracking activities are definitely not negligible, as well as radar sensors. Besides, for correlation purposes it is clear that the most recent orbital elements shall be used and thus it does not make sense to aim at two days of delay in the absence of technical limitations.
The scientific gap that this work intends to fill is not clear. In the introduction, explicit reference is made to the identification, shape, material, and size of the space objects, but they are not tackled in the manuscript. This work present a method to correlate optical observations with TLEs, which is not novel nor aligned to the state of the art and does not provide any scientific contribution to the existing literature. The success rate and efficiency mentioned in the paper are not clearly defined and the results highly depend on the scenario considered.
The results show a very poor performance and the need to use relatively high threshold values, compared to typical sensor measurement noise, very likely due to the decision of using outdated TLEs. The recommendation provided in the conclusions of selecting TLE data spanning 4 days for correlation does not seem correct and definitely not aligned to other studies. The evaluation of the performance of the algorithm is evaluated with a single satellite, thus ignoring correlation ambiguity that may happen when more than one target are observed.
Comments on the Quality of English LanguageMany of the terms used in the manuscript are not aligned to the space surveillance and tracking methodology and in fact sound weird. Some examples: "management of space target catalogues", "catalogue library", "establishment of cataloguing libraries".
Round 2
Reviewer 3 Report
Comments and Suggestions for Authors
The main concerns of the previous review remain there:
The scientific gap that this work intends to fill is not clear. In the introduction, explicit reference is made to the identification, shape, material, and size of the space objects, but they are not tackled in the manuscript. This work present a method to correlate optical observations with TLEs, which is not novel nor aligned to the state of the art and does not provide any scientific contribution to the existing literature. The success rate and efficiency mentioned in the paper are not clearly defined and the results highly depend on the scenario considered.
Comments on the Quality of English LanguageSome terms have been fixed but still doesn't sound too natural.
Author Response
Dear reviewer 3,
We have meticulously taken your comments into account and sincerely sorry for disturbing you again while you were reviewing our article. Based on your comments, we have made corrected modifications on the re-revised manuscript. For the responses, please see the attachment.
Thank you very much for your attention and time. Look forward to hearing from you.
Sincerely,
Jiannan Sun
28 March,2025
Changchun Observatory, National Astronomical Observatory, Chinese Academy of Sciences
